# Postvaccination SARS-CoV-2 Infections among Healthcare Professionals: A Real World Evidence Study

**DOI:** 10.3390/vaccines10040511

**Published:** 2022-03-25

**Authors:** Alessandro Perrella, Sara Mucherino, Ilaria Guarino, Mariagiovanna Nerilli, Alberto Enrico Maraolo, Nicolina Capoluongo, Enrico Coscioni, Ugo Trama, Enrica Menditto, Valentina Orlando

**Affiliations:** 1UOC Emerging Infectious Disease with High Contagiousness, AORN Ospedali dei Colli P.O. C Cotugno, 80131 Naples, Italy; mariagiovannanerilli@hotmail.it (M.N.); alberto.maraolo@ospedalideicolli.it (A.E.M.); nicolina.capoluongo@ospedalideicolli.it (N.C.); 2CIRFF (Center of Drug Utilization and Pharmacoeconomics), Department of Pharmacy, University of Naples Federico II, 80131 Naples, Italy; sara.mucherino@unina.it (S.M.); ilaria.guarino@cirff.it (I.G.); enrica.menditto@unina.it (E.M.); 3Regional Task Force COVID-19, Campania Region, 80143 Naples, Italy; enrico.coscioni@regione.campania.it (E.C.); ugo.trama@regione.campania.it (U.T.); 4Division of Cardiac Surgery, AOU San Giovanni di Dio e Ruggi d’Aragona, 84131 Salerno, Italy; 5Directorate-General for Health Protection, Campania Region, 80143 Naples, Italy

**Keywords:** COVID-19, SARS-CoV-2, vaccination, healthcare professionals, healthcare workers, COVID-19 symptoms

## Abstract

Due to the COVID-19 pandemic, all countries with a global mobilization started to produce and authorize vaccines, prioritizing healthcare professionals (HCPs) to reduce transmission. The aim of this study was to assess post-vaccination infections’ occurrence among HCPs and their correlation with symptom onset. A retrospective cohort study was carried out in the Campania Region from December 2020 to April 2021. Data were retrieved from the Regional Health Information System of the Campania Region (Sinfonia). The study cohort included subjects that had all received at least one vaccine dose. Risk ratios (RRs) adjusted for age and sex (95% confidence intervals) were performed to assess differences in the prevalence between HCPs who tested positive or negative for COVID-19. Univariate and multivariate logistic regression models were used to evaluate the association between symptoms and vaccination status. Findings revealed that HCPs had a lower risk of contracting COVID-19 after receiving at least one vaccine dose, and this risk decreased with age. Furthermore, not having full vaccination coverage may predict a severe/critical evolution of the disease. This study provides a snapshot of the initial state of the Italian vaccination campaign on HCPs. A surveillance approach using Big Data matched to clinical conditions could offer a real analysis in the categorization of subjects most at risk.

## 1. Introduction

Since the World Health Organization (WHO) declared the emergence of the coronavirus disease 2019 (COVID-19) pandemic on 11 March 2020, over five million people have died worldwide, [1] including over 130,000 people in Italy [2]. COVID-19 is a clinical syndrome correlated to infection resulting from the novel severe acute respiratory syndrome coronavirus-2 (SARS-CoV-2) and is characterized by the involvement of several organs with different outcomes. The WHO recognized a clinical spectrum of SARS-CoV-2-related symptoms, ranging from: asymptomatic or paucysymptomatic infection, such as patients without symptoms that are consistent with COVID-19; mild infection, such as minor flu-like symptoms (e.g., fever, cough, sore throat, malaise, headache, muscle pain, nausea, vomiting, diarrhea, loss of taste and smell); severe infection causing acute respiratory distress syndrome (ARDS) and pneumonia; to critical infection (e.g., respiratory failure, septic shock, and/or multiple organ dysfunction) [2,3].

In that context, all healthcare professionals (HCPs) were engaged at an early stage to manage and treat, to the best of scientific and clinical practice knowledge, patients with COVID-19, and were therefore constantly exposed, even if using disposable personal protection equipment.

Due to the unprecedented impact on the healthcare systems of all countries, global resources were mobilized to find a cure or to develop a vaccine against the SARS-CoV-2 [3,4,5,6].

Despite the fact that the development of vaccines for human use normally requires several years and major expenditure, several vaccines were produced and authorized between the first and second waves of the pandemic based on different technologies. In Italy, at the end of 2020 and throughout 2021, in accordance with the European Medical Agency (EMA) and the Italian Medicine Agency (AIFA), a number of vaccines were authorized: the BNT162b2 mRNA (Pfizer-BioNTech) [7,8] and ChAdOx1 nCoV-19 adenoviral (AZD1222; Oxford-AstraZeneca) [9], mRNA (Moderna) [10,11], and Adenoviral AD06 (J&J) [12,13]. Typical side effects common to all these vaccines include pain at the injection site, fever, fatigue, headache, muscle pain, chills, and diarrhea. Because the vaccines are based on different technologies, the chances of any of these side effects occurring after vaccination differ according to the specific vaccine [2]. According to the WHO, less common side effects reported for some COVID-19 vaccines have included severe allergic reactions such as anaphylaxis (an extremely rare reaction).

Large-scale vaccination of at-risk groups and, subsequently, of the general population proved to be the single most effective public health measure to mitigate the coronavirus pandemic [14]. During the SARS-CoV and MERS-CoV outbreaks, history taught us that patient-to-patient and patient-to-healthcare worker transmission occurred mostly in healthcare settings [15,16,17]. Since the level of risk of nosocomial transmission to healthcare professionals (HCPs) was elevated, the national vaccination campaign prioritized HCPs to reduce transmission. Indeed, the effectiveness of the Italian vaccine campaign to control the COVID-19 disease, as with others, was not merely dependent on vaccine efficacy and safety but also on early vaccine acceptance among healthcare workers at the end of December 2020, which appeared to have a decisive role in the successful control of the pandemic [18]. Unfortunately, SARS-CoV-2 infection after vaccination has been shown to be able to reoccur, as vaccines to prevent COVID-19 disease do not offer 100% protection [19,20,21,22]. Hence, even those who had received one or two doses of the vaccine began to become infected, initially with milder symptoms. According to WHO considerations, it is widely recognized that, after vaccination, it usually takes a few weeks for the body to build immunity against SARS-CoV-2, the virus that causes COVID-19 [2]. Then, there is the possibility that a person could be infected with SARS-CoV-2 just before or after vaccination and still become sick with COVID-19. This is because the vaccine has not yet had enough time to provide protection [2]. In this scenario, the aim of the present study was to prospectively assess the occurrence of post-vaccination infections, after second pandemic wave, among healthcare professionals (HCPs) in the Campania Region and its correlation with the onset of disease symptoms from the beginning of the vaccination campaign to April 2021.

## 2. Materials and Methods

### 2.1. Study Design and Population

A retrospective cohort study was carried out among healthcare professionals (HCPs) vaccinated against COVID-19 in the Campania Region from 27 December 2020 to 15 April 2021. The study cohort included subjects who had all received at least one vaccination of the BNT162b2 mRNA vaccine (Pfizer-BioNTech). Subjects who tested positive for COVID-19 before receiving the first vaccine dose were excluded from the analysis. Then, the study cohort was divided as follows:–Cohort 1: HCP uninfected after COVID-19 vaccination named “COVID-19 Negative”;–Cohort 2: HCP infected after COVID-19 vaccination named “COVID-19 Positive”.

Furthermore, Cohort 2 was analyzed and divided into subgroups according to vaccination status: (i) HCPs who received at least one dose of vaccine; (ii) HCPs who received two doses of vaccine; (iii) HCPs who received two doses of vaccine and had the 15-day time frame needed to provide effective protection, referred to hereafter as an ‘effective dose’.

### 2.2. Data and Sample Collection

Healthcare workers were tested with nasopharyngeal swabs, which were collected by trained personnel from the Regional Healthcare system and/or authorized and trained territorial laboratory staff. RT-PCR testing was performed with the use of a standardized RT-PCR machine from the Coronavirus Network Laboratory (CoroNetLab), using four gene analysis RdRP, S, and N genes specific to SARS-CoV-2, and the E gene with results expressed as the cycle threshold (Ct). A Ct value of less than 30, which indicated an increased viral load, was used to determine infectivity [23,24]. The samples tested were considered fully positive in cases where all 4 genes were amplified by RT-PCR, whereas in all other cases, the results were considered doubtful, and the tests were repeated. In this case, patient consent was required and given for the release of all SARS-CoV-2 PCR test results before or after vaccination. All positive subjects were followed up until their first negative PCR test. Clinical symptoms were collected in accordance with the National Institute of Health for all nasal swab positive individuals. Typical COVID-19 symptoms were fever, cough, or change in or loss of taste or smell. Subjects were recorded as having other symptoms if they reported any of the following: shortness of breath, sore throat, runny nose, headache, muscle aches, extreme fatigue, diarrhea, nausea or vomiting, or small itchy red patches on fingers or toes, on the follow-up questionnaire with a symptom onset date within 14 days before or after the PCR positive sample date. Data extraction was carried out monthly via the Sinfonia Data source to obtain regular reports of vaccine/positive trends from December 2020 to April 2021.

### 2.3. Data Source

The data source used for the study was the Regional Health Information System of the Campania Region, officially named ‘Sinfonia’ (https://sinfonia.soresa.it/sinfonia/ Accessed on: 1 June 2021), includes information on patient demographics of about 6 million residents in a southern Italian region (Campania) and comprising a well-defined population (about 10% of the whole national population). The database incorporates a data management system that has already been validated and described in previous studies [25,26,27,28,29]. All data are gathered within Sinfonia in an encrypted and anonymized form in accordance with current privacy regulations. Hence, the analyses were carried out using transparent data encryption protocols. The legal owner of the original data is the Local Health Unit (LHU).

All subjects engaged in a healthcare procedure or testing for COVID-19, including HCPs, signed informed written consent to include their data in the Local Health Unit Database and therefore in Sinfonia. Data extracted in this study were related to the positive and negative status of individuals according to their vaccine schedule and time elapsed.

During the pandemic outbreak, the Sinfonia database was implemented, containing all COVID-19-related information and all infected patients’ data and related clinical history (symptoms, hospital admission and related follow-up, previous clinical status) in accordance with European Privacy Policy to manage the pandemic in order to create a tool to support national health governance in managing the extraordinary emergency. The aims of the Sinfonia tool [30] were to (i) apply data science methods to Big Data in order to assess pandemic trends; (ii) create predictive algorithms through artificial intelligence (AI) methods; and (ii) use machine learning (ML) methods using the Python scripting model (Spyder IDE 64bit version, Massachusetts Institute of Technology, Cambridge, MA, USA) to perform predictive analysis on virus contagiousness. Characteristics of Sinfonia are described in Appendix A.

### 2.4. Statistical Analysis

The baseline characteristics of HCPs who received at least one COVID-19 vaccine dose were analyzed using descriptive statistics. Quantitative variables were described as counts and percentages. Chi-square and *t*-test were performed to determine the difference between vaccinated HCPs tested positive for COVID-19 and those who tested negative. Crude and age-adjusted prevalence rates were calculated. The difference in prevalence between infected and uninfected HCPs after vaccination was assessed overall and stratified by age group, expressed as risk ratios (RR) with 95% confidence intervals (CI), to investigate the correlation between infection and sample age. Standardization was performed using a direct method whereby the Italian population on 1 January 2020 was used as the standard population.
$$\begin{array}{c} \mathrm{Directly}\;\mathrm{Standardized}\;\mathrm{Rate}\\ \frac{\sum_{i=l}^{m}w_{i}\cdot t_{i}}{\sum {}_{i=l}^{m}}\cdot k \end{array}$$
where (*Ti = ni/n*) = rate in stratum ‘*i*’ of the study population; *ni* = number of cases in stratum ‘*i*’ of the study population; N = size of the study population in stratum ‘*i*’; *wi* = size of stratum ‘*i*’ of the reference population; *m* = number of considered strata; and *k* = multiplicative constant. Univariate and multivariate logistic regression models were additionally carried out to evaluate the association between the clinical spectrum (or disease-related symptoms) and vaccination status, such as one vaccine dose received, two vaccine doses received, or an effective dose received, i.e., two vaccine doses plus the 15 days needed to provide full protection. The models were adjusted for age and sex in order to mitigate confounding variables in the analysis. A *p*-value of <0.05 was considered statistically significant. Data management was performed with SQL server v2018 (Microsoft, Redmond, WA, USA). Analyses were carried out with SPSS v17.1 (IBM, Armonk, NY, USA).

## 3. Results

Between 27 December 2020 and 15 April 2021, 285,149 healthcare professionals aged 18–75 years living in Campania Region received at least one dose of BNT162b2 mRNA vaccine (Pfizer-BioNTech) and did not test COVID-19 positive before receiving their first vaccine dose.

As shown in Figure 1, about 99% (282,055) of the HCPs involved in the analysis tested negative for COVID-19, whereas 1% (3094) tested positive. Of those infected, 53.2% (1647) were vaccinated with one dose; 13.0% (403) received two doses; and 33.7% (1044) were fully vaccinated with an effective dose.

Considering the baseline sample characteristics (Table 1), the analysis showed that among the total cohort of HPCs vaccinated, 40.6% were aged between 41 and 60 years, 35.9% were over 60 years, and only 2.4% were under 40 years. Furthermore, the percentage of HCPs over 60 years old who tested COVID-19 negative was higher than for those aged under 60 years (99.2% 0–40 y vs. 98.7% 41–60 y).

Figure 2 shows the estimation of vaccination as a protective factor in the onset of COVID-19 through RR values. Overall, the risk ratio of testing positive after receiving at least one vaccine dose was 0.012 (95% CI: 0.011–0.014). Stratification by age group also showed that the risk ratio of testing positive after receiving at least one vaccine dose was slightly lower among HCPs aged over 60 years (RR: 0.008; 95% CI: 0.008–0.009).

The results in Table 2 show the symptoms and duration of COVID-19 in post-vaccination infected HCPs. Information on disease symptoms was not available for 442 subjects (14.4%) included in the analysis. For those where it was available, the percentage of subjects with COVID-19 symptoms decreased in the cohort of those vaccinated with an effective dose compared to those vaccinated with an ineffective dose (one or two vaccine doses received). Indeed, among 98 HCPs with paucysymptomatic symptoms, only 1.1% were vaccinated with an effective dose. Furthermore, among 85 HCPs with mild symptoms, 5.6% received only one or two vaccine doses, and 1.6% were vaccinated with an effective dose. Altogether, only 0.13% developed severe or critical symptoms of the disease, and they corresponded to those who had received a single dose of vaccine. In addition, for HCPs infected after vaccination, the mean number of days from the first positive test to the first negative test was lower in those vaccinated with an effective dose (10 ± 9 days).

Finally, univariate and multivariate logistic regression analyses revealed that vaccination status was a predictor of contracting symptomatic COVID-19. The proportion of HPCs who received one vaccine dose were at almost two times higher risk of the onset of symptoms than those who received an effective dose (Table 3).

## 4. Discussion

According to evidence from the literature, to the best of our knowledge, this study provides a snapshot of the initial state of the Italian vaccination campaign in the Campania Region, which began by prioritizing healthcare professionals. Therefore, the findings of this retrospective study confirm what has been published in the literature to date. Namely, it was shown that the first tranche of HCPs covered by the vaccination campaign had a lower risk of contracting COVID-19 disease after receiving at least one vaccine dose. To prove our point, a recent study confirmed that the effectiveness of a COVID-19 vaccination program strongly depends on the vaccination rate and the efficacy of the vaccine [31]. Moreover, a recent systematic review analyzing 13 studies related to COVID-19 vaccine efficacy confirmed that most of the vaccines on the market between the first and second pandemic waves appear to be effective and safe [32].

Another important finding from this prospective study is purely age-related. In this regard, the analyses revealed that older HCPs (particularly above 60 years) had a lower risk of contracting COVID-19 disease after receiving the vaccination. This can be explained in several ways. First, it is likely that older HCPs in a particular emergency may have been less engaged on the frontline due to their hypothetical fragility caused by concomitant diseases. Secondly, older HCPs may generally hold management positions that prevent their constant presence in the hospital ward. Other studies have already investigated specific cases of HCPs infection [33,34]. Particularly, another Italian study carried out in Turin supported our hypothesis by revealing that most of the infected healthcare workers were those in direct contact with patients, whereas the administrative staff members infected were significantly lower [35].

Although the vaccination coverage, regardless of the type of vaccine technology administered, is not 100% effective, the proportion of vaccinated HCPs who contracted COVID-19 disease was relatively low (1.1%). Therefore, we found that not having a full vaccination coverage may predict a severe/critical evolution of COVID-19 disease. The evidence available so far has already revealed that a double-dose vaccination is generally recommended [36,37,38,39]. There are several studies in the literature comparing the effects of single-dose and double-dose vaccination confirming these hypothesis and demonstrating that a double-dose vaccination produces a stronger immune response than single-dose vaccination [36,37,38,39]. Corroborating our findings, Ebinger et al. examined the response to the Pfizer-BioNTech mRNA vaccine in a large cohort of healthcare workers, including those with and without prior COVID-19 infection, observing that the antibody response following a single vaccine dose in HCPs who had recovered from confirmed prior COVID-19 infection was similar to the antibody response following two doses of vaccine in persons without prior infection [40]. Moreover, an Israeli study confirmed that HCPs who received a single dose of vaccine were less likely to be asymptomatic [14]. Therefore, in real-life scenarios, healthcare professionals should not avoid considering post-vaccination symptoms as vaccine-related but should promptly test for COVID-19 [14].

This study has several limitations to be taken into account. First, even though we provide one of the most extensive documentations of a cohort of vaccinated HCPs, the number of cases that resulted in infection were relatively small. Second, this cohort represented mostly adults (over 40 years); thus, we could not determine if severe/critical infection cases could be due to coexisting illnesses or simply age, since we did not have data on clinical conditions and comorbidities. Third, most of the vaccinated HCPs who then tested positive for COVID-19 were under 60 years, which could be a further confounding factor in the assessment of the real impact of severe/critical symptoms. To overcome this issue and rule out that the sample age could be a confounding factor, the logistic regression models were adjusted for age and sex. Fourth, we may have missed asymptomatic cases, despite the intensive effort to test all exposed health care workers during surveillance, that could have presented asymptomatic infection prior to the testing period and were therefore negative once they underwent screening. This factor may have led to an underestimation of the difference in infected and uninfected vaccinated HCPs. Finally, in many cases, the peri-infection antibody titer that was available had been obtained on the day of detection of the infection (which, in some cases, could have been a few days into the infection period) and therefore was possibly already elevated because of the infection. However, since most cases were detected in the pre-symptomatic stage, we expect that this contamination of results was minor. Moreover, we found that among the cases for whom both peri-infection and earlier neutralizing antibody results were available, the majority of titers were lower during the peri-infection period than during the earlier period, which also suggests that this contamination was negligible. If such contaminations were substantial, the result would likely be biased toward the null hypothesis of no relationship between antibody titers and breakthrough infection.

## 5. Conclusions

COVID-19 vaccines were introduced in an emergency situation and were supported by several clinical trial results. There were no clinically relevant symptoms of COVID-19 in the majority of those who benefited from the vaccination campaign after the second pandemic wave. However, in our experience, COVID-19 vaccines, firstly administrated to Italian healthcare professionals (HCPs), clearly decreased the risk of contracting COVID-19 disease. On the other hand, the few infected HCPs who fully completed vaccination coverage against COVID-19 revealed significantly lower risk of developing a worse clinical outcome. Although carried out in the early phase of the vaccination strategy, the present real-world study provides noteworthy findings useful for encouraging vaccination campaigns against COVID-19 and in any pandemic situation. The findings certainly encourage a surveillance approach based on the use of integrated Big Data systems for all clinical conditions, as this could offer precise and real analysis with a low incidence of errors in the categorization of subjects most at risk.

## Figures and Tables

**Figure 1 vaccines-10-00511-f001:**
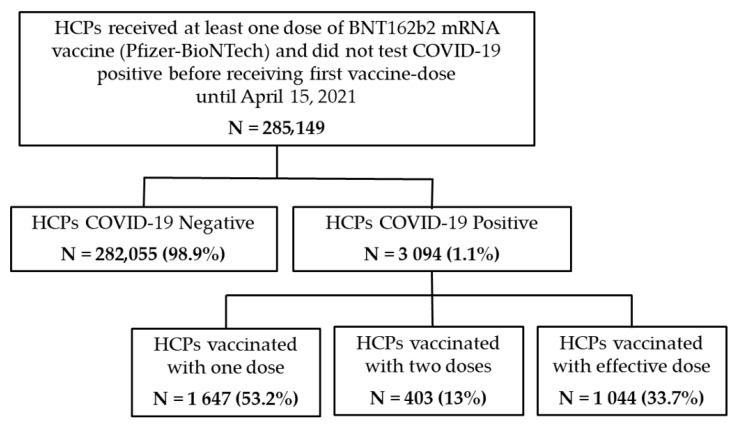
Flowchart.

**Figure 2 vaccines-10-00511-f002:**
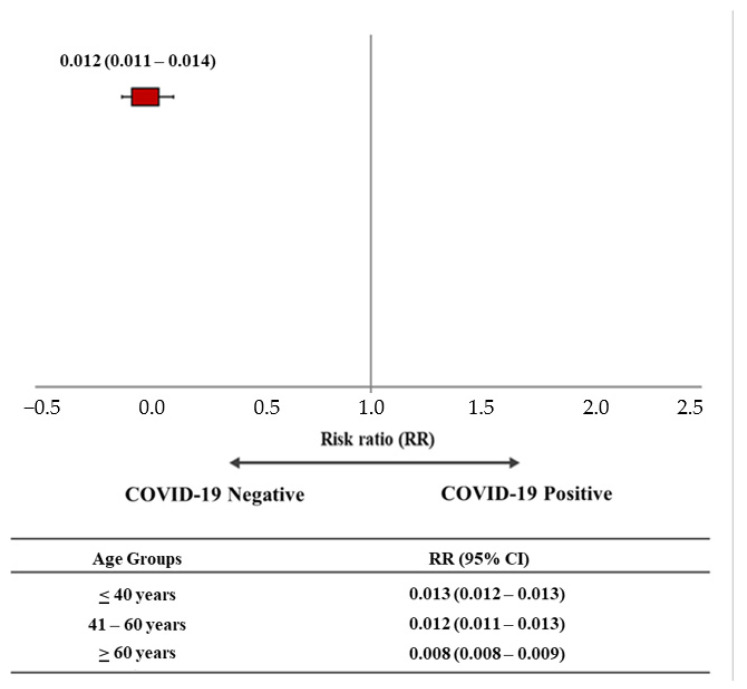
Risk of testing positive after receiving at least one COVID-19 vaccine dose stratified by age.

**Table 1 vaccines-10-00511-t001:** Characteristics of healthcare professionals received at least one COVID-19-vaccine dose.

	COVID-19 Negative	COVID-19 Positive	Total
	N (%)	N (%)	N (%)
Total	282,055 (98.9)	3094 (1.1)	285,149
Sex			
Male	140,137 (98.9)	1497 (1.1)	141,634 (49.7)
Female	141,918 (98.8)	1597 (1.2)	143,515 (50.3)
Mean Age (±SD)	52 (±14)	55 (±94)	52 (±17)
Age Groups			
≤40 years	66,121 (98.7)	899 (1.3)	67,020 (2.4)
41–60 years	114,413 (98.7)	1372 (1.3)	115,785 (40.6)
≥60 years	101,521 (99.2)	823 (0.8)	102,344 (35.9)

**Table 2 vaccines-10-00511-t002:** Disease symptoms and duration of COVID-19 in post-vaccination infected HCPs.

Symptoms and Duration of COVID-19	Vaccination Status N (%)			Overall N (%)
	1st Vaccine Dose	2nd Vaccine Dose	Effective Dose *	
	1647 (53.2)	403 (13.0)	1044 (33.7)	3094
Disease Symptoms (%) °				
Asymptomatic	1368 (83.1)	354 (87.8)	744 (71.3)	2466 (79.7)
Paucysinintomatic	77 (4.7)	9 (2.2)	12 (1.1)	98 (3.2)
Mild	60 (3.6)	8 (2.0)	17 (1.6)	85 (2.7)
Severe	2 (0.1)	-	-	2 (0.1)
Critical	1 (0.1)	-	-	1 (0.03)
Days from 1st positive test to 1st negative test (Mean; ±SD)	13 (±9)	11 (±8)	10 (±9)	12 (±8)

* Effective dose: two vaccine doses plus 15 days. ° Percentage calculated from the total cohort tested COVID-19 positive excluding patients with unavailable disease symptoms.

**Table 3 vaccines-10-00511-t003:** Predictors of symptom severity onset among COVID-19 positive HCPs.

Vaccination Status	OR	95% CI	*p*-Value °
Effective dose *	Reference	Reference	Reference
1st vaccine dose	2.303	1.535–3.455	0.001
2nd vaccine dose	1.133	0.324–1.763	0.654

* Effective dose: two vaccine doses plus 15 days. ° *p*-value less than 0.05 was considered to be statistically significant.

## Data Availability

Permission use anonymized data to this study was granted to the researchers of Regional Direction for Health Management, Pharmaceutical Unit. Requests for information on data access can be directed to Perrella A. and Orlando V.

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
