# Peer review of "Postvaccination SARS-CoV-2 Infections among Healthcare Professionals: A Real World Evidence Study"

_vaccines, 2022, doi:10.3390/vaccines10040511_

Round 1

Reviewer 1 Report

Review Report of Vaccines 

Title: Post vaccination SARS-CoV-2 Infections among Healthcare

         Professionals: a Real World Evidence Study.

Comments to the Authors:

  1. The discussion section (Introduction) in the present form is relatively weak and should be strengthened with more details and justifications about COVID-19 pandemic.
  2. Authors have used 1st vaccine dose, 2nd vaccine dose as well as effective dose of COVID-19 pandemic.  Please provide the COVID-19 Booster dose status and compare the results with the previous one.
  3. Authors should discuss: what are the side effects of COVID-19 vaccines?
  4. Please polish the grammar.

Please discuss the conclusion section of the article in more details.

Author Response

Response to Reviewer 1 Comments

Point 1: The discussion section (Introduction) in the present form is relatively weak and should be strengthened with more details and justifications about COVID-19 pandemic.

Response 1: Dear reviewer, on behalf of all co-authors, thank you for your suggestions to improve the manuscript. Indeed, we have implemented and argued the introductory section in greater depth. Particularly, we placed our study in greater context by providing a more complete overview of the first-wave pandemic situation. We have defined, according to the WHO, the symptomatology and Clinical Spectrum of SARS-CoV-2 Infection, and we have stressed the focus of our study, namely the health care professionals (HCPs) who were the first to receive vaccination in the very first vaccination campaign.

Point 2: Authors have used 1st vaccine dose, 2nd vaccine dose as well as effective dose of COVID-19 pandemic.  Please provide the COVID-19 Booster dose status and compare the results with the previous one.

Response 2: As mentioned in the material and methods, this study was managed and retrospectively evaluated on a cohort of HCPs after second wave and therefore  third dose was not already applied to these subjects.

Point 3: Authors should discuss: what are the side effects of COVID-19 vaccines?

Response 3: We have discussed in the Introduction section the most common side effects resulting from the administration of COVI-19 vaccines, even though they are based on different technologies. Please, see Lines 57-63.

Point 4: Please polish the grammar.

Response 4: Thanks for suggestion. We carried out a complete revision of the manuscript grammar and improved its form.

Point 5: Please discuss the conclusion section of the article in more details.

Response 5: We have enriched the conclusions section with the major findings of our research and the future steps to be expected in this field.

Reviewer 2 Report

Estimated Authors of the paper

"Postvaccination SARS-CoV-2 Infections among Healthcare Professionals: a Real World Evidence Study", as an occupational physician I've read your paper with great interest.

In fact, this "real-world" study sheds some lights on the actual impact of SARS-CoV-2 vaccination in the healthcare settings. Briefly, your study suggests that receiving SARS-CoV-2 vaccine reduced (please note the past of this statement) the odds for developing severe COVID-19 in HCW from Campania Region. This effect depended on the number of vaccine doses received by HCWs, consistently with other reports. Also the risk for developing SARS-CoV-2 infection (See Figure 2) was substantially reduced.

Despite its potential interest, the present article is affected by several shortcomings, some of them could be addressed by Authors by means of a more extensive discussion. Unfortunately, other may require some extensive reworking of this study.

1) the selection of the study population is quite unclear; according to your text, you recruited a total of 285,149 HCWs, whose clinical features were retrieved from the Regional Health Information System of Campania Region; Authors should discuss in the methods section how they obtained the right and the institutional authorization for using corresponding data. Was the study preventively waived by an Ethical Comittee?

2) the sample is characterized by substantial differences, as Authors should stress that individuals tested positive were substantially younger than those tested negative (Table 1, age groups chi squared test 127.0, equals to p < 0.001); it has obvious consequences on the results and their interpretation, and it should be discussed;

3) Authors reports some RR estimates in Figure 2, but it is unclear how were they calculated; if I've correctly assessed the test, it is the RR for being SARS-CoV-2 in a certain age group in vaccinated vs. the very same occurrence in the very same age group in not vaccinated. As the detailed information on the age groups by vaccination status and infection status is not directly provided, Authors should amend their text accordingly; I would suggest to remove Figure 2 by replacing it with a more detailed Table including the aforementioned information.

4) The statement "Univariate and multivariate logistic regression models were carried out to evaluate the association between severe disease symptoms and status of vaccination (one vaccine dose, two vaccine dose vs effective dose)." should be reworked by including a more extensive and detailed description of the statistical models that were implemented.

Author Response

Response to Reviewer 2 Comments

Point 1: "Postvaccination SARS-CoV-2 Infections among Healthcare Professionals: a Real World Evidence Study", as an occupational physician I've read your paper with great interest. In fact, this "real-world" study sheds some lights on the actual impact of SARS-CoV-2 vaccination in the healthcare settings. Briefly, your study suggests that receiving SARS-CoV-2 vaccine reduced (please note the past of this statement) the odds for developing severe COVID-19 in HCW from Campania Region. This effect depended on the number of vaccine doses received by HCWs, consistently with other reports. Also the risk for developing SARS-CoV-2 infection (See Figure 2) was substantially reduced. Despite its potential interest, the present article is affected by several shortcomings, some of them could be addressed by Authors by means of a more extensive discussion. Unfortunately, other may require some extensive reworking of this study.

Response 1: Dear Reviewer, on behalf of all the authors, we really appreciate your interest in the present study. We understand your point of view. We have therefore made a general revision of the manuscript by implementing the introduction, discussion and conclusion sections providing a new version of the paper.

Point 2: The selection of the study population is quite unclear; according to your text, you recruited a total of 285,149 HCWs, whose clinical features were retrieved from the Regional Health Information System of Campania Region;

Response 2: Thanks for your question. SINFONIA is a BigData repository of all clinical data including HCPs. All individual once underwent to healthcare procedure, including vaccine, sign informed written consent. From this DB, we extracted only positive and negative nasal swab status according to vaccine. Further we calculated time elapsed form complete o incomplete vaccination and positive nasal swab for SARS-COV-2.

Point 3: Authors should discuss in the methods section how they obtained the right and the institutional authorization for using corresponding data. Was the study preventively waived by an Ethical Comittee?

Response 3: Totally understand this point. To carry out this study, all patients’ data were fully anonymized and were analyzed retrospectively. Indeed, for this type of study, formal consent was not required according current national established by the Italian Medicines Agency, and according to the Italian Data Protection Authority, neither ethical committee approval nor informed consent was required for our study. We provided to submit the study to the Ethical Committee of “Aziende Ospedaliere di Rilievo Nazionale e di Alta Specializzazione – A.Cardarelli/Santobono – Pausilipon” which confirmed and certified that this type of study did not require any Approval. This aspect is widely discussed in the ‘Institutional Review Board Statement’ section.

Point 4: The sample is characterized by substantial differences, as Authors should stress that individuals tested positive were substantially younger than those tested negative (Table 1, age groups chi squared test 127.0, equals to p < 0.001); it has obvious consequences on the results and their interpretation, and it should be discussed;

Response 4: We understand your perplexity. As you stated and noted in Table 1, most of the HCPs vaccinated and then tested positive for COVID-19 were under 60 years. During our analyses we considered that this sample aspect could be a confounding factor compared to not developing severe/critical symptoms as for the youngest age. To overcome this issue and to avoid that the sample age could be a confounding factor, the logistic regression models were adjusted for age and sex. We provided to clarify this issue in the study limitations list in the discussion section.

Point 5: Authors reports some RR estimates in Figure 2, but it is unclear how were they calculated; if I've correctly assessed the test, it is the RR for being SARS-CoV-2 in a certain age group in vaccinated vs. the very same occurrence in the very same age group in not vaccinated. As the detailed information on the age groups by vaccination status and infection status is not directly provided, Authors should amend their text accordingly; I would suggest to remove Figure 2 by replacing it with a more detailed Table including the aforementioned information.

Response 5: We understand your question. Actually, the three aspects you mention, i.e. age, vaccination status and the risk of contracting COVID-19 post vaccination, were assessed in the preliminary analysis by building logistic regression models (unpublished material). As mentioned above, the results of these analyses were not significant, as age was a confounder in the model analysis. For this reason, the logistic regression model was adjusted for age and sex to avoid statistical confounding (Table 3). We therefore assessed the age variable individually and calculated the relative risk of contracting COVID-19 post-vaccination. For this reason, Figure 2 shows age as the only variable. In addition, Figure 2 gives an overall risk ratio value, which is then stratified by age in the table below. We have specified this issue further in the manuscript.

Point 6: The statement "Univariate and multivariate logistic regression models were carried out to evaluate the association between severe disease symptoms and status of vaccination (one vaccine dose, two vaccine dose vs effective dose)." should be reworked by including a more extensive and detailed description of the statistical models that were implemented.

Response 6: We provided to better explain the model both in the methods and results sections. 

Round 2

Reviewer 1 Report

NA

Reviewer 2 Report

My formal concerns were properly addressed; therefore, I've no further requests and I'm endorsing the acceptance of this paper.